# Fertility and Pregnancy Outcomes after Fertility-Sparing Surgery for Early-Stage Borderline Ovarian Tumors and Epithelial Ovarian Cancer: A Single-Center Study

**DOI:** 10.3390/cancers15225327

**Published:** 2023-11-08

**Authors:** Mu-En Ko, Yi-Heng Lin, Kuan-Ju Huang, Wen-Chun Chang, Bor-Ching Sheu

**Affiliations:** 1Department of Obstetrics and Gynecology, National Taiwan University Hospital Yunlin Branch, Yunlin 640, Taiwan; y07550@ms1.ylh.gov.tw (M.-E.K.); y06058@ms1.ylh.gov.tw (K.-J.H.); 2Department of Obstetrics and Gynecology, National Taiwan University Hospital, Taipei 100, Taiwan; 112023@ntuh.gov.tw (Y.-H.L.); bcsheu@ntu.edu.tw (B.-C.S.)

**Keywords:** fertility sparing surgery, epithelial ovarian cancer, ovarian cancer, borderline ovarian tumor, fertility, pregnancy, childbirth

## Abstract

**Simple Summary:**

Although fertility sparing surgery (FSS) may be a favorable surgical option for young women with early-stage borderline ovarian tumors (BOTs) or epithelial ovarian cancer (EOC) who wish to preserve their fertility, there still possessed risks of recurrence and disease progression during follow-up. The purpose of this retrospective article is to investigate the treatment outcomes in early-stage BOTs/EOC patients who underwent FSS. In this cohort, we found that the rate of pregnancy was higher among the married BOTs patients who had no prior live birth experience than among those who had prior live birth experience. The fertility and pregnancy outcome presented in this article may provide an important reference during pre-operative counseling.

**Abstract:**

This study examined treatment outcomes, including preserved fertility, menstrual regularity, and pregnancy outcomes, in patients with stage I epithelial ovarian cancer (EOC) or borderline ovarian tumors (BOTs) who underwent fertility-sparing surgery (FSS). Patients with stage I EOC and BOTs who were aged 18–45 years and underwent FSS between 2007 and 2022 were retrospectively reviewed. Significant differences between various subgroups in terms of disease recurrence, menstrual irregularity due to the disease, and pregnancy outcomes were analyzed. A total of 71 patients with BOTs and 33 patients with EOC were included. In the BOT group, the median age was 30 (range, 19–44) years. Recurrence occurred in eight patients, with one case exhibiting a malignant transformation into mucinous EOC. Among the 35 married patients with BOTs, 20 successfully conceived, resulting in 23 live births and 3 spontaneous abortions. A higher pregnancy rate was observed in those without prior childbirth (82.4%) than in those who had prior childbirth (33.3%). In the EOC group, the median age was 34 (range, 22–42) years. Recurrence occurred in one patient. Menstrual regularity was maintained in 69.7% of the patients. Among the 14 married patients in this group, 12 achieved a total of 15 pregnancies (including 2 twin pregnancies), 16 live births, and 1 spontaneous abortion. The results of the study confirmed that FSS is a favorable surgical option for young women with early-stage BOTs or EOC who wish to preserve their fertility. However, additional investigations are needed to validate these findings.

## 1. Introduction

Ovarian cancer is a highly common malignancy [1] and has the highest mortality rate among all cancers of the female genital tract [2]. Most cases of ovarian cancer are of epithelial ovarian cancer (EOC), and over 70% of EOC diagnoses are made at an advanced stage [3]. Approximately 15% of all patients with EOC present with localized early-stage cancer; in these patients, the cure rate after optimal treatment is 90% [3]. Despite the predominance of EOC in older women, particularly postmenopausal women, approximately 10–13% of all EOC cases are diagnosed in women aged <40 years [4,5]. Borderline ovarian tumors (BOTs), a distinct condition from EOC, are characterized by epithelial proliferation and nuclear atypia without invasive stromal destruction [6,7]. Furthermore, BOTs typically occur in younger women and are often diagnosed at earlier stages. Approximately 75–80% of all BOT cases are detected in their early stage, with one-third of these cases occurring in women aged <40 years [8]. The prognosis for BOTs is considerably more favorable than that for EOC with a 5-year overall survival (OS) rate of >90% [9].

Many women in industrialized countries choose to delay childbirth, to their 30s and even 40s, as more participate in the workforce and pursue higher education [10,11]. Therefore, preserving the fertility of young patients with EOC or BOTs while ensuring optimal oncological outcomes through fertility-sparing surgery (FSS) is a crucial topic that warrants further investigation.

The guidelines of the European Society of Medical Oncology recommend comprehensive FSS (cFSS) for young patients with any stage IA histology or stage IC1/IC2 with favorable histology (i.e., low-grade tumors) [12]. The recent National Comprehensive Cancer Network guidelines recommend FSS for patients who wish to preserve their fertility despite having apparent early-stage disease, low-risk tumors such as early-stage invasive epithelial tumors and low malignant potential tumors (ovarian borderline epithelial tumors), or a combination of both. For patients with stage IA or IB EOC who wish to preserve their fertility, a suitable treatment option may involve comprehensive surgical staging along with unilateral salpingo-oophorectomy (USO) for stage IA EOC or bilateral salpingo-oophorectomy (BSO) for stage IB EOC. For patients with BOTs who wish to sustain their fertility, either USO or BSO combined with residual disease resection may be considered.

This study evaluated several treatment outcomes, namely preserved fertility, menstrual regularity, and pregnancy outcomes, in patients undergoing FSS for EOC or BOT.

## 2. Methods

This retrospective study was conducted at a single tertiary hospital in Taiwan between January 2007 and September 2022. The study protocol was approved by the Research Ethics Committee of National Taiwan University Hospital before the initiation of data collection. The medical records of patients who underwent FSS at our hospital for EOC or BOTs were retrospectively reviewed. Patients were eligible for inclusion in this study if they (1) were aged between 18 and 45 years at the time of initial diagnosis and expressed a desire to preserve their fertility while having a comprehensive understanding of the risks and benefits of FSS, (2) were pathologically confirmed to have stage I disease (BOT or EOC), and (3) underwent initial FSS and regular follow-up and received adjuvant chemotherapy at our hospital. Patients were excluded if they (1) had nonepithelial histology (i.e., sex cord–stromal tumor and germ cell tumor), (2) had incomplete medical records, and (3) did not undergo regular follow-up. 

All included patients had undergone either simple FSS (sFSS) or comprehensive FSS (cFSS). FSS is defined as any surgical procedure that preserves the uterus and at least one ovary; under this definition, FSS is still defined as such regardless of whether it is conducted for unilateral or bilateral disease. This can be accomplished by performing salpingo-oophorectomy or unilateral oophorocystectomy while preserving at least one ovary and the uterus. 

sFSS refers to FSS performed without any additional surgical staging procedure. By contrast, cFSS includes not only FSS but also surgical staging procedures, such as peritoneal evaluation (peritoneal washing cytology, omentectomy/omental biopsy, and peritoneal biopsy if necessary) and lymph node evaluation, if deemed necessary. Lymph node evaluation involves the dissection or sampling of lymph nodes or the identification and removal of enlarged lymph nodes through palpation. 

In this study, pathologic reviews were conducted by an gynecologic pathologist. Tumor stage was determined using the International Federation of Gynecology and Obstetrics (FIGO) classification framework. Patients with EOC included in this study may have had three to six postoperative cycles of platinum-based adjuvant chemotherapy based on their risk factor profiles. 

Follow-up assessments comprised physical examinations, ultrasonography, and serum cancer antigen (CA)-125 level measurements. These assessments were conducted every 1–3 months during the first 2 years following primary treatment, every 3–6 months during the subsequent 3 years, and then annually thereafter. In patients with suspected disease recurrence, computed tomography or magnetic resonance imaging was performed to detect potential local recurrence, pelvic lymphadenopathy, or distant metastasis. The total follow-up duration was recorded for each patient. 

The following patient characteristics were analyzed: age, body mass index, preoperative serum CA-125 level, tumor histological type, tumor size, FSS comprehensiveness, FIGO stage, adjuvant chemotherapy history, obstetric and gynecological history, underlying medical conditions, marital status, menstrual regularity, development of persistent amenorrhea (defined as secondary amenorrhea lasting for a minimum of 6 months after primary treatment), numbers of pregnancies and deliveries after FSS, type of conception (namely, natural conception or assisted reproductive technology (ART)-assisted conception), and adverse pregnancy events (e.g., preterm delivery). In addition, data regarding the interval between primary treatment and recurrence, recurrence site, management strategies for recurrence, pathologic findings pertaining to recurrent lesions, and mortality were analyzed. 

### Statistical Analysis

Statistical analyses were performed using SAS (SAS software for Windows version 9.4, SAS Institute, Cary, NC, USA). Demographic data were summarized using descriptive statistics. Pearson’s chi-square tests and Fisher’s exact tests were performed to identify significant differences between various subgroups in terms of disease recurrence and menstrual irregularity due to disease. A *p* value of <0.05 indicated statistical significance.

## 3. Results

### 3.1. Study Cohort

Initially, the medical records of 244 patients who underwent FSS for EOC or BOTs were retrospectively reviewed. Subsequently, 84 of them were excluded because they had nonepithelial or non-BOT tumors, In addition, 21 patients with EOC and 35 patients with BOTs were excluded because of an advanced FIGO stage or the lack of complete medical records or regular follow-up. Finally, the data of 33 patients with EOC and 71 patients with BOTs were analyzed (Figure 1).

### 3.2. Clinicopathological Characteristics of Included Patients

The clinicopathological characteristics of all the included patients are summarized in Table 1. The median age of the patients with BOTs was 30 (range, 19–44) years. Their median follow-up duration was 71 (range, 6–152) months with 87.3% (62/71) of them having a follow-up duration of >24 months. The most common histologic type among the patients with BOTs was mucinous BOTs (73.2%). Of these patients, approximately 5% had stage IA disease, 42.2% had stage IC disease, and 2.8% had stage IB disease. Approximately 62.5% of all patients with BOTs underwent sFSS either through salpingo-oophorectomy or through oophorocystectomy, and the remaining 37.5% underwent cFSS. Because of the presence of suspicious malignant cells in their ascites, two patients (2.82%) received adjuvant chemotherapy after a thorough consultation. 

The median age of the patients with EOC was 34 (range, 22–42) years. Their median follow-up duration was 97 (range, 3–180) months with 87.9% of them (22/33) having a follow-up duration of >24 months. The most common histology type was mucinous EOC. Of these patients, 45.5% had mucinous EOC and 30% had a clear cell type. Approximately 54.5% of all patients with EOC had stage IC disease and 45.5% had stage IA disease; however, no patient had pathologically confirmed bilateral disease (IB). Approximately 57.6% of the patients underwent cFSS, and 42.4% underwent sFSS. Frozen tissue samples (intraoperatively collected) were unavailable for 5 of the 14 patients who underwent sFSS; the remaining six patients had a benign histology or BOTs. Approximately 60.6% (20/33) of all the patients with EOC received 3–6 cycles of adjuvant chemotherapy. 

### 3.3. Tumor Recurrence

Among the patients with BOTs, the recurrence rate was 11.27% (8/71). Notably, the recurrence of BOTs was significantly associated with the FIGO stage, histological type, and preoperative CA-125 level (Table 2); however, no association was noted between the recurrence of BOTs and the comprehensiveness of FSS or inclusion of cystectomy in the FSS. Of the eight patients who experienced BOT recurrence, seven were successfully treated through surgery. One patient with mucinous borderline tumor had disease recurrence that eventually progressed to mucinous EOC. Despite undergoing debulking surgery and receiving salvage chemotherapy, she died within 11 months after disease progression.

Among the patients with EOC, only one patient had disease recurrence after 21 months of regular follow-up. She subsequently sought help from other health-care providers and was thus lost to follow-up. None of the risk factors examined seem to be related to recurrence of the EOC group (Appendix A).

### 3.4. Menstrual Regularity

Menstrual regularity was maintained in 69.7% (23/33) of all the patients with EOC. Among these, five patients aged between 36 and 44 years experienced amenorrhea before reaching the age of 45 years; all of these patients had received adjuvant chemotherapy. These five patients exhibited persistent secondary amenorrhea for 7 (immediately after primary treatment), 13 (developed 8 years after treatment, at the age of 43 years), 16 (occurred 7 years after primary treatment, at the age of 42 years), 31 (occurred 8 years after primary treatment, at the age of 38 years), and 75 (occurred 5 years after primary treatment, at the age of 44 years) months, respectively. Menstrual regularity was maintained in 97.2% (69/71) of all patients with BOTs. Two patients experienced amenorrhea after undergoing debulking surgery following tumor recurrence. 

In patients with EOC and those with BOTs, none of the analyzed potential risk factors (e.g., FIGO stage and FSS comprehensiveness) were found to be associated with menstrual irregularity (Appendix A). 

### 3.5. Pregnancy Outcomes

In both the BOT and EOC study groups, none of the unmarried patients achieved childbirth or attempted to conceive. Among the patients with BOTs, 49.3% of them were married (35/71), with 23 being already married before undergoing treatment and remaining married throughout the follow-up period and 12 getting married during the follow-up period. Among the patients with EOC, 42.4% of them (14/33) were married, with seven already being married before undergoing treatment and remaining married throughout the follow-up period and seven getting married during the follow-up period (Table 3).

Among the patients with BOTs, 20 patients conceived, resulting in 23 live births and 3 spontaneous abortions. The overall pregnancy rate was 57.1% (20/35) with a corresponding live birth rate of 88.5% (23/26). Among the 18 patients with BOTs who were already married before undergoing surgery and had prior live-birth experiences, only 33.3% (6/18) conceived after treatment, resulting in 6 live births and 1 spontaneous abortion. By contrast, 3 of the 5 patients who were already married but had no prior live birth experience conceived and 11 of the 12 patients who got married after treatment conceived, resulting in a total of 19 pregnancies and 17 live births (Table 4). The median interval between the last menstrual cycle before the first post-treatment pregnancy and the primary treatment was 39 (range, 1–114) months. Most of the patients conceived naturally. Only one patient received ART treatment. One preterm singleton birth at a gestational age of 35 weeks was recorded due to preterm labor.

Among all eight patients who experienced recurrence, five of them achieved a total of seven pregnancies, resulting in five live births and two spontaneous abortions (Table 5). BOT1 had one live birth before recurrence. BOT2 did not become pregnant before recurrence and underwent comprehensive staging surgery after 37 months of follow-up. BOT3-5 all conceived naturally, and each had one live birth after experiencing recurrence. BOT6 achieved three pregnancies through in vitro fertilization (IVF), resulting in one live birth and two spontaneous abortions. BOT7 initially underwent right salpingo-ophorectomy and omental biopsy, did not become pregnant after 3 years of follow-up, and then received left salpingo-ophorectomy after consultation. BOT8 received a second fertility-sparing surgery with partial oophorectomy but did not seek to achieve pregnancy due to her unmarried status.

Among the patients with EOC, 12 patients conceived, resulting in a total of 15 pregnancies, which eventually led to 16 live births and 1 spontaneous abortion, resulting in an overall pregnancy rate of 85.7% (12/14) and a corresponding live birth rate of 93.3% (14/15; Table 6). The median interval between the last menstrual cycle before the first post-treatment pregnancy and the primary treatment was 31 (range, 9–84) months. Three patients received ART treatment. Two preterm births were recorded, both of which were twin pregnancies delivered by cesarean section due to preterm labor at gestational ages of 31 and 34 weeks.

## 4. Discussion

A previous study reported that the rate of first childbirth among women aged >35 years increased by 23% between 2000 and 2014 [13]. Because delayed childbirth has become a global trend [10,11], FSS should be considered for patients with early-stage EOC or BOTs who wish to preserve their fertility. Thus, the safety of FSS as a treatment for early-stage EOC and BOTs is a crucial topic. 

Kajiyama et al. reported no significant difference in OS or recurrence-free survival between patients with early-stage EOC undergoing FSS and those undergoing radical surgery [14]. Watanabe et al. indicated that FSS is an acceptable treatment option for stage IA and IC1 EOC, highlighting the favorable reproductive outcomes associated with this treatment [15]. A meta-analysis revealed no difference in OS or recurrence-free survival between patients with stage I EOC who underwent FSS and those who underwent radical surgery [16]. In a multi-institutional study, laparoscopic FSS was proposed as a viable option for stage I EOC [17]. In their study involving patients with stage IC2/IC3 EOC, Nasioudis et al. concluded that FSS was not associated with poor OS [18]. In our study, only one patient experienced disease recurrence, which occurred after 21 months of regular follow-up. No significant correlation was noted between recurrence and the patient characteristics analyzed in the present study. Although our EOC study group had a similar patient composition in terms of stage IC group percentage (54.5% vs. 57.5%), histology distribution, and adjuvant chemotherapy percentage (60.6% vs. 63.4%), the rate of tumor recurrence (3% vs. 14.7%) in our EOC study group was lower than that reported in a previous study by Kajiyama et al. [14]. This difference in recurrence rates may be attributed to our relatively small sample size.

FSS is currently a widely accepted strategy for the treatment of patients with BOTs who wish to preserve their fertility. This procedure is typically performed for early-stage BOTs. However, the feasibility of FSS in advanced BOTs remains a topic of debate. An analysis of data retrieved from a Danish national database revealed a 5-year DFS rate of 97.6% for patients with stage I BOTs who underwent FSS; this rate is similar to the 5-year DFS rate for patients with BOTs who underwent conventional surgery [19]. One study indicated FSS to be a feasible option even for patients with advanced BOTs, reporting that FSS was not associated with an increased risk of relapse in young patients [20]. Lu et al. also reported findings supporting the efficacy of FSS in terms of fertility outcomes and overall prognosis in advanced BOTs [21]. However, whether minimal invasive surgery should be used in FSS remains controversial. In a study conducted in France, laparoscopic FSS for BOTs was found to be associated with an elevated cyst rupture risk, incomplete staging, and an increased recurrence rate [22]. However, Kasaven et al. determined no significant difference in the recurrence rate among the types or routes of FSS [23]. 

The rate of recurrence in our BOT cohort was 11.27% (8/71), which is similar to those reported in prior studies [19,23]. Recurrence in our BOT group was significantly associated with disease stage (*p* = 0.0354), histology type (*p* = 0.0179), and preoperative CA-125 level (*p* = 0.0238) but not FSS comprehensiveness and the inclusion of cystectomy. In our study, two of the five patients with bilateral BOTs experienced recurrence; both patients presented with IC disease and underwent cFSS as primary treatment. A study focusing on bilateral disease revealed no difference in OS between patients undergoing FSS and those undergoing radical surgery, although the FSS group exhibited poorer DFS than did the radical surgery group; the study also reported similar recurrence and DFS rates between patients undergoing bilateral ovarian cystectomy and those undergoing USO with contralateral cystectomy in the FSS group [24]. 

Because recurrence may occur in patients with BOTs while they are still young and wishing to conceive someday, a second FSS may be performed after thorough consultation. Wang et al. confirmed that a second FSS was as safe as radical surgery for recurrent BOTs with a pregnancy rate of 46.9% and a live birth rate of 81.3% [25]. In our study group, 8 of 71 patients with BOTs experienced tumor recurrence. Of these, two received radical surgery: one as a result of choice and the other due to malignant transformation. Among the six patients who received a second FSS, four achieved six pregnancies, including four live births and two spontaneous abortions after the surgery.

Five (15.1%) of our patients with EOC experienced amenorrhea before 45 years of age after primary treatment, which is a higher rate compared with that reported in the study by Satoh et al., where only a 5% (6/121) persistent secondary amenorrhea rate in their EOC group was reported [26]. Because we did not have data on antral follicle counts or anti-Mullerian hormone levels to evaluate the difference in baseline ovarian reserve, a younger median age in their patient group (29 vs. 34) may explain the difference in amenorrhea rates. 

Because the act of conception is affected by not only biological determinants but also social and emotional ones, patients undergoing FSS may not necessarily desire to conceive. Studies have indicated that only 16–65.4% of all patients with early-stage EOC who underwent FSS reported active attempts to conceive [27,28], which may be explained by their fear of disease recurrence [29] and marital status [30]. Chen et al. observed the most common reasons for patients who did not attempt conception after FSS were either being unmarried (70%) or already having children (15%) [31]. A review study reported that among patients with EOC who underwent FSS and actively attempted to conceive, the rate of pregnancy was 79% (242/307) and that of live births was 76–96% [32]. In our study, 42.4% of the patients with EOC (14/33) were married, and 85.7% of these married patients (12/14) had at least one post-treatment pregnancy, resulting in a live birth rate of 93.3% (14/15). Unlike in patients with BOTs, patients with EOC had similar pregnancy rates (100% vs. 80%, respectively) regardless of whether they had prior live birth experience.

Among patients with BOTs who underwent FSS, the rate of pregnancy varied widely (17.9% to 100%) [32]. Among patients with BOTs who expressed a desire to conceive, an overall pregnancy rate of 32–88% was reported for all stages [33,34]. Helpman et al. reported a pregnancy rate of 85.6% (18/21) among patients with advanced stage BOTs who underwent FSS and had at least one pregnancy; the rate of live births in these patients was 76.4% (26/34) [19]. In our study, only 49.3% of the patients with BOTs were married (35/71). Among these married patients, 57.1% (20/35) had at least one pregnancy; the corresponding rate of live births was 88.5% (23/26). A subgroup analysis revealed a higher pregnancy rate in patients without prior live birth experience (82.4% (14/17)) than in those with prior live birth experience (33.3% (6/18)).

For patients with BOTs and EOC who underwent FSS, ART should be considered if persistent infertility is noted. A systematic review addressing this issue concluded that ART can be initiated in patients with stage I BOTs [34] if required. A systematic review suggested that in EOC, ART may not be associated with an increased risk of relapse, and subsequent pregnancies do not lead to poor oncological outcomes [35]. 

Marital status and prior childbirth are correlated with pregnancy rates and the desire to conceive after FSS, as demonstrated in this study and previous studies [29,30,31]. Therefore, a thorough preoperative consultation is essential for confirming whether a patient has a strong desire to conceive. The patient’s age, ovarian reserves, marital status, marital plans, and live birth experience should be considered in the consultation. To help patients make informed surgical decisions, clinicians should provide information regarding the percentage of patients who attempted to conceive after treatment, the corresponding pregnancy rates, live birth rates, and recurrence risks after undergoing FSS relative to those in standard staging surgery, as reported in previous studies. This information can help patients understand the risks and the odds of success and make well-informed choices. 

## 5. Limitations

This study has some limitations. Because this was a retrospective study conducted in a single tertiary hospital, biases may be present in the findings. Notably, the small number of patients in each subgroup might have limited the statistical power of our findings. In addition, the exclusion of 10 patients with early-stage EOC and 21 patients with early-stage BOTs due to loss of follow-up may have affected the final results. Because we used marital status as an indicator of desire for pregnancy without verifying the patients’ actual attempt to become pregnant due to the study design and IRB approval, the results regarding actual pregnancy rates and live birth rates require further verification.

## 6. Conclusions

FSS may be a favorable surgical option for young women with early-stage BOTs or EOC who wish to preserve their fertility. However, the risks of recurrence and disease progression during follow-up cannot be eliminated completely. We determined that the rate of pregnancy was higher among the married BOTs patients who had no prior live birth experience than among those who had prior live birth experience. Therefore, during a preoperative consultation, physicians must consider their patient’s age, ovarian reserves, marital status, marital plans, and prior live birth experience to tailor surgical decisions to her specific needs.

## Figures and Tables

**Figure 1 cancers-15-05327-f001:**
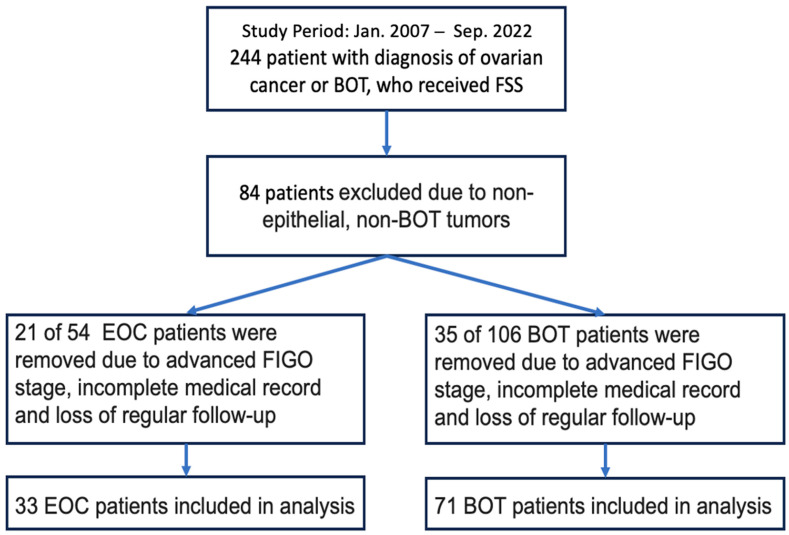
Flowchart for patient selection.

**Table 1 cancers-15-05327-t001:** Clinicopathological characteristics of all included patients.

	Epithelial Ovarian Cancer	Borderline Ovarian Tumors
Total	33	71
Median age (years)	34 (22–42)	30 (19–44)
Median follow-up interval (month)	97 (3–180)	71 (6–152)
Follow-up interval ≥ 24 months (%)	87.9 (29/33)	87.3 (62/71)
FIGO stage		
IA (%)	45.5 (15/33)	55.0 (39/71)
IB (%)	-	2.8 (2/71)
IC (%)	54.5 (18/33)	42.2 (30/71)
Histology		
Mucinous (%)	45.5 (15/33)	73.2 (52/71)
Serous (%)	3.0 (1/33)	21.1(15/71)
Clear Cell (%)	30.3 (10/33)	-
Endometrioid (%)	21.2 (7/33)	-
Seromucinous (%)	-	5.6 (4/71)
Lesion site		
Right (%)	18.2 (6/33)	19.7 (14/71)
Left (%)	81.8 (27/33)	73.2 (52/71)
Bilateral (%)	-	7.1(5/71)
Diameter		
Median diameter (cm)	16 (3–40)	15 (5–50)
<10 cm (%)	33.3 (11/33)	21.1 (15/71)
≥10 cm (%)	66.7 (22/33)	78.9 (56/71)
Preoperative CA-125 > 35 IU/mL (%)	51.5 (17/33)	54.9 (39/71)
Surgical approach		
Laparoscopy (%)	15.2 (5/33)	33.8 (24/71)
Laparotomy (%)	84.8 (28/33)	66.2 (47/71)
Comprehensiveness of surgery		
Comprehensive FSS (%)	57.6 (19/33)	38.0 (27/71)
Simple adnexectomy/cystectomy (%)	42.4 (14/33)	62.0 (44/71)
Intraoperative rupture (%)	36.4 (12/33)	30.1 (22/71)
Lymphadenectomy (%)	54.5 (18/33)	28.2 (20/71)
Omentectomy (%)	63.6 (21/33)	35.2 (25/71)
Adjuvant chemotherapy (%)	60.6 (20/33)	4.2 (3/71)
Recurrence (%)	3.0 (1/33)	11.3 (8/71)
Death (%)	-	1.4 (1/71)

**Table 2 cancers-15-05327-t002:** Analysis of risk factors for recurrence in patients with borderline ovarian tumors.

	Recurrence (%)				*p* Value
FIGO stage	Ia	Ib		Ic	0.0354
	2.63(1/38)	0 (0/2)		22.58 (7/31)	
Histology	Mucinous	Serous		Seromucinous	0.0179
	5.77 (3/52)	33.33 (5/15)		0 (0/4)	
Preoperative CA-125	<35 U/mL	35–100 U/mL		>100 U/mL	0.0238
	3.13 (1/32)	15.79 (3/19)		20 (4/20)	
Comprehensiveness of FSS	sFSS		cFSS		0.7014
	13.64 (6/44)		7.41 (2/27)		
Extent of ovarian surgery in FSS	Salpingo-oophorectomy only		Inclusion of cystectomy		0.189
	7.55 (4/53)		22.22 (4/18)		
Lymphadenectomy	Yes		No		0.4267
	5 (1/20)		13.73 (7/51)		
Omentectomy	Yes		No		0.8033
	12 (3/25)		10.87 (5/46)		
Adjuvant chemotherapy	Yes		No		-
	0 (0/3)		11.76 (8/68)		

**Table 3 cancers-15-05327-t003:** Marital status of patients.

	Remaining Married	Getting Married	Divorced	Single
Epithelial Ovarian Cancer (n = 33)	7	7	1	18
Borderline Ovarian Tumors (n = 71)	23	12	2	34

**Table 4 cancers-15-05327-t004:** Pregnancy outcomes in patients with borderline ovarian tumors.

	Borderline Ovarian Tumor						
	Number of Patients	Number of Patients Achieving Pregnancy	Number of Pregnancies	Live Birth	Abortion	Preterm	ART *
Married with prior childbirth	18	6	7	6	1	0	0
Married without prior childbirth	5	3	6	4	2	0	1
Married after treatment	12	11	13	13	0	1 **	0

* ART: assisted reproductive technology. ** Singleton with preterm labor delivered at 35 gestational weeks.

**Table 5 cancers-15-05327-t005:** Clinical characteristics of patients with recurrent disease.

	Age	FIGO Stage	FSS Procedure	Histology Type	Recurrence (Month)	Treatment	Prior Live Birth	Achieved Pregnancy	Achieved Live Birth
EOC1	35	IC1	LSO + BPLNS + omentectomy	Clear cell	21	Loss of follow-up	0	0	0
BOT1	33	IC2	RSO	Mucinous	23 *	Debulking surgery and C/T	1	1	1
BOT2	35	IC1	LSO + appendectomy + omental biopsy + LPLNS	Mucinous	37 M	Complete staging	0	0	0
BOT3	25	IA	LSC RSO	Serous	70 M	Left cystectomy	0	1	1
BOT4	22	IC1	LSO	Serous	71 M	LSC RPO	0	1	1
BOT5	23	IC1	LSO	Serous	66 M	RSO	0	1	1
BOT6	35	IC1	LSC left cystectomy	Mucinous	17 M	LSO	0	3	1
BOT7	28	IC2	RSO + omental biopsy	Serous	36 M	LSO	0	0	0
BOT8	29	IC3	LSO + Right cystectomy + omentectomy	Serous	69 M	RPO	0	0	0

* 23 months: BOT; 32 months: malignant transformation to invasive mucinous EOC; 51 months: expired. C/T: salvage chemotherapy. RSO/RPO/LSO: right salpino-oophorectomy/right partial oophorectomy/left salpingo-oophorectomy. BPLNS/LPLNS: bilateral pelvic lymph node sampling/left pelvic lymph node sampling.

**Table 6 cancers-15-05327-t006:** Pregnancy outcomes in patients with epithelial ovarian cancer.

	Epithelial Ovarian Cancer						
	Number of Patients	Number of Patients Achieving Pregnancy	Number of Pregnancies	Live Birth	Abortion	Preterm	ART *
Married with prior childbirth	4	4	4	4	0	0	0
Married without prior childbirth	3	2	2	2	0	0	1
Married after treatment	7	6	9	10	1	2 **	2

* ART: artificial reproductive technique. ** MCDA twin with preterm labor underwent cesarean section at 31 gestational weeks. DCDA twin with preterm labor underwent cesarean section at 34 gestational weeks.

## Data Availability

Data is unavailable due to institution-specific privacy restrictions.

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
