# Peer review of "Fertility and Pregnancy Outcomes after Fertility-Sparing Surgery for Early-Stage Borderline Ovarian Tumors and Epithelial Ovarian Cancer: A Single-Center Study"

_cancers, 2023, doi:10.3390/cancers15225327_

Round 1

Reviewer 1 Report

Comments and Suggestions for Authors

Thank you to let me review this interesting paper ! 

Major comments

Methods : Please detail how the data were collected : into medical files ?

Methods : please detail if patients were all followed in the university hospital or were some of them followed in other hospitals or private practices. In this case, how were the data collected ?

Methods : Data collection on pregnancies should be detailed : questionnaire to the patient (with a possible recall bias) ? obstetrical departement files ?

In the whole manuscripts, authors take into account the marital status. However, some couples may wish to have a child without being married. Therefore, is it possible to know as well patients being in a relatioship ?

Methods and results : Do you have any data whether women have tried to be pregnant or not ?

Results : Could you precise the proportion of women excluded because of advanced FIGO separetly from those excluded because a lack of complete medical records ?

Results : page 3 lines 110 and 118-119 : are the proportions of each histological type representative of usual histological types in taiwanese women ? (as this distribution of histological types is not the same in some other continents) : this should be discussed in the discussion to show external validity.

Results page 4 line 134 : precise that this patient had a pregnancy before recurrence. Could you also precise the delay between pregnancy and recurrence ? Was this pregnancy obtained naturally or with assisted reproductive technology ?

Results : page 5 lines 145 to 147 : concernng the 5 patients who became amnorrheic, how long was the follow-up after the end of chemotherapy ? (as some transitory chemotherapy-induced ameniorrhea can occur without being definitive). The term « amenorrheic » should also be explained in the methods section (12 months ?).

Results : page 5 lines 161-165 : if available, please add data on women who tried to be pregnant

Results page 6 : if possible, please add data on means of conception : naturally ? ART ?

Results : page 6 lines 166-168 Do you have data on time to conception (which is an indirect marker of fertility) ?

Results : did any patient benefit from oocyte vitrification of any other pertiflity preservation method at management of after treatment ?

Discussion page 7 lines 239-240 : If you have no data on women who tried to be pregnant without success (infertile women), this point needs to be adressed in the discussion as nulliparity after FSS is not necessarly due to a wrong selection of motivated patients to reach pregnancy.

Discussion page 7 241 the counseling before FSS should also include age and the ovarian reserve evaluation of the patient, as pregnancy rates might be lower in older patients.

Discussion : authors should also discuss the opportunity to be referred to a reproductive medicine specialist after treatmene to discuss oocyte vitrification after FSS (if the pregnancy project is not at short term).

Limitations : recall bias should also be discussed depending on the mean to collect data on pregnancies and pregnancies outcomes.

Limitations : depending on the number of women lost for follow-up, please discuss this bias.

Limitations : if no data on pregnancy attempt : please add this point in the limitations

Minor comments

Methods Page 2 line 67 : rephrase «and lymph nodal evaluation » into « and if needed lymph nodal evaluation » (as it is not necessary in borderline ovarian tumors)

Results page 5 line 155 : change « were married during the follow-up » into « got married » or « became married »

Discussion : line 180 : reference 13 concerns epidemiological data in the united states. Is this transposable to taiwanese women ? Please prefer if available a reference concerning taiwanese women.

Comments on the Quality of English Language

Minor editing of English language required

Author Response

Dear Expert Reviewer in the field, our reply is in the attached files 

Thank you and sincerely,

Mu-En Ko and Wen-Chun Chang

Reviewer 2 Report

Comments and Suggestions for Authors

Ko and coworkers present the results of a retrospective single-center study conducted on patients affected by Early-Stage Borderline Ovarian Tumors or Epithelial Ovarian Cancer and treated with Fertility-Sparing Surgery from 1 January 2007 to 30 September 2022.

Other authors and meta-analyses have addressed the same topic, drawing the same conclusions, albeit separately, for the two types of tumors. The study therefore does not present any innovative information.

The manuscript then seems roughly prepared as if the authors had presented a draft and not the final text.

Main shortcomings:

1. The Abstract is missing.

2. The references are written inconsistently even if the instructions for authors specify that “Your references may be in any style, provided that you use the consistent formatting throughout.”

3. On page 3, in the part relating to “3.2. Clinicopathological Characteristics of the Included Patients”, the numbers reported as results for BOT patients do not correspond in many cases to those reported in Table 1.

4. There is no reference to Table 3 in the text of the results.

5. On page 6, in the part relating to “3.5. Pregnancy Outcomes”, the numbers reported as results for BOT patients (see line 166) do not correspond to those reported in table 5.

Comments on the Quality of English Language

Authors should use terms more appropriate to the data they want to present, for example:

1. Table 5 and table 6, why write “Patients achieved pregnancy” and not “Number of patients achieving pregnancy” and why write “Achieved pregnancy” and not “Number of pregnancies”?

2. Why talk about marital status and prior childbirth and not about the desire for pregnancy?

Author Response

Dear Expert Reviewer in the field, our reply is the attached file.

Thank you and sincerely,

Mu-En Ko and Wen-Chun Chang

Reviewer 3 Report

Comments and Suggestions for Authors

This is quite a large series of patients with early-stage borderline and invasive EOC treated with fertility-sparing surgery from a single institution. It is generally well written and well referenced. I have the following comments:

(1) In Line 59, you state that sFSS means "simple fertility-sparing surgery" but need to explain what the "c" stands for in cFSS

(2) Line 64. It is not possible to know the stage if no surgical staging has been performed. Patients undergoing sFSS should be considered to have "Clinical" stage 1 disease, as opposed to "Surgical" stage 1 for patients who have undergone surgical staging.

(3) Line 133 and 139. You state that some patients with both BOTs and EOCs had a recurrence, but could you please indicate where the recurrence was. For example, was it in the contralateral ovary, omentum, lymph node etc

(4) Table 2. Why were patients with stage 1 BOT given chemotherapy, and is there a reason why there is no p value given for Adjuvant chemotherapy?

(5) Table 4. You are assuming that marital status equates with desire for fertility. In order to determine whether or not treatment caused infertility, you need to indicate how many patients were trying to conceive, regardless of whether or not they were married or had a prior live birth. Similarly, in your "Conclusion", you state that "physicians must consider their patient's marital status, marital plans, and live birth experience to tailor surgical decisions...' These factors are all irrelevant to the decision making. The only factor that is relevant is the patient's desire for future fertility.

(6) Lines 188 and 213. What does the term "radical" mean? Does it mean surgical staging or TAH/BSO, with or without surgical staging?

Comments on the Quality of English Language

The grammar is generally good, and needs only minor editing.

Author Response

Dear Expert Reviewer in the field, our reply is in the attached file.

Thank you and sincerely,

Mu-En Ko and Wen-Chun Chang

Round 2

Reviewer 2 Report

Comments and Suggestions for Authors

The revision of the manuscript may have improved the scientific content, but the improvement cannot be properly appreciated due to poor use of the English language by the authors.

Comments on the Quality of English Language

I strongly suggest that authors have their manuscript's use of English completely reviewed by an expert native English speaker.

Author Response

Dear Reviewer,

Thank you for your comments and we do value your suggestion on our prior English editing. Revised version of the manuscript was conducted by two external expert native English speakers. Please let us know if you have further questions. 

Thank you and Sincerely

Dr. Wen-Chun Chang

Round 3

Reviewer 2 Report

Comments and Suggestions for Authors

The revision of the text has now made the manuscript worthy of publication.